# Review of Electrothermal Micromirrors

**DOI:** 10.3390/mi13030429

**Published:** 2022-03-10

**Authors:** Yue Tang, Jianhua Li, Lixin Xu, Jeong-Bong Lee, Huikai Xie

**Affiliations:** 1School of Mechatronical Engineering, Beijing Institute of Technology, Beijing 100081, China; tangyue@bit.edu.cn; 2Beijing Engineering Research Center of Detection and Application for Weak Magnetic Field, Department of Physics, University of Science and Technology Beijing, Beijing 100083, China; jianhuali@ustb.edu.cn; 3Department of Electrical and Computer Engineering, The University of Texas at Dallas, Richardson, TX 75080, USA; jblee@utdallas.edu; 4School of Integrated Circuits and Electronics, Beijing Institute of Technology, Beijing 100081, China; 5BIT Chongqing Center for Microelectronics and Microsystems, Chongqing 400030, China

**Keywords:** MEMS, electrothermal micromirrors, electrothermal microactuators, thermal bimorphs, chevron microactuators, thermo-pneumatic microactuators, thermo-capillary microactuators

## Abstract

Electrothermal micromirrors have become an important type of micromirrors due to their large angular scanning range and large linear motion. Typically, electrothermal micromirrors do not have a torsional bar, so they can easily generate linear motion. In this paper, electrothermal micromirrors based on different thermal actuators are reviewed, and also the mechanisms of those actuators are analyzed, including U-shape, chevron, thermo-pneumatic, thermo-capillary and thermal bimorph-based actuation. Special attention is given to bimorph based-electrothermal micromirrors due to their versatility in tip-tilt-piston motion. The exemplified applications of each type of electrothermal micromirrors are also presented. Moreover, electrothermal micromirrors integrated with electromagnetic or electrostatic actuators are introduced.

## 1. Introduction

In recent decades, MEMS mirrors, or micromirrors, have been drawing extensive attentions in consumer electronics and optical industry for a wide range of applications, such as projectors, optical switching, LiDAR, face recognition, biomedical imaging, and spectroscopy [1,2,3,4,5,6]. There are primarily four types of micromirrors, i.e., electromagnetic, electrostatic, piezoelectric and electrothermal micromirrors. Electromagnetic micromirrors can achieve high speed scanning and large driving force, but external magnets are typically needed, which will cause the form factor relatively large [7,8]. Electrostatic micromirrors can realize high speed scanning with low power consumption but need high drive voltage [9]. Piezoelectric micromirrors consume low power and requires only moderately high driving voltage but have limited scanning range [10]. Electrothermal micromirrors typically have large scan range and low drive voltage but consume relatively high power [5,11]. So, every type of micromirrors has their advantages and disadvantages. This paper is focused on electrothermal micromirrors.

In general, electrothermal micromirrors, according to their electrothermal actuation mechanisms, can be divided into hot-cold arm, chevron, thermal-bimorph, thermo-pneumatic, and thermocapillary micromirrors. Hot-cold arm and chevron actuators can generate in-plane forces which can create in-plane deflection [12,13]. With a sophisticated hinge structure, these two types of actuators also can be made for out-of-plane deflection [14,15]. A thermal bimorph actuator consists of two stacked materials with different coefficients of thermal expansions (CTEs). The free side of a bimorph cantilever can achieve a large out-of-plane deflection when changing the bimorph’s temperature [16]. Thermocapillary and thermo-pneumatic actuators have also been employed in scanning micromirrors [17,18]. Furthermore, electrothermal actuators combined with electrostatic or electromagnetic actuators have been explored to take their respective advantages while overcome their drawbacks [19,20,21].

This paper is organized as follows. Section 2 introduces the general thermal effects that are commonly used for electrothermal actuation. Section 3 is focused on the electrothermal actuation mechanisms, including U-shape and V-shape actuation, thermo-pneumatic actuation, thermo-capillary actuation, and thermal bimorph-based actuation; their working principles are reviewed and their characteristics are analyzed and compared. Section 4 is a compressive review of electrothermal micromirrors based on the actuation mechanisms mentioned above.

## 2. Thermal Effects for Actuation

Thermal actuation is an easy way for converting electric energy to mechanical energy without the need of complicated structures. There are several thermal effects that can be used for thermomechanical actuation, including thermal expansion of a solid, temperature-dependent contact angle of a liquid droplet on a solid surface, and thermal expansion of a gas, as shown in Figure 1.

Thus, thermomechanical actuation can be primarily divided into three categories: solid-based thermal actuation, liquid-based thermal actuation, and gas-based thermal actuation. Among them, solid-based thermal actuation can be divided into single-material type and bi-material (or bimorph) type. Single-material solid thermal actuators further include hot-cold arm (U-shaped) actuators and chevron (V-shaped) actuators. Gas-based thermal actuation is primarily thermo-pneumatic actuation. These actuation types are summarized in Figure 2. The principles and characteristics of these actuation mechanisms are discussed in Section 3 in details.

## 3. Electrothermal Action Mechanisms

### 3.1. Solid-Based Thermal Actuation

#### 3.1.1. Single-Material Solid Thermal Actuators

##### Hot-Cold Arm Actuators

A hot-cold arm actuator (also named U-shaped actuator) consists of a cold arm and a hot arm. These two arms typically have the same thickness but different widths, and they are connected at the free end [22], as shown in Figure 3a. Since the resistance in a narrow arm is bigger than that of a wide one, the temperature of the narrow arm becomes higher than that of the wide one when the actuator is injected with an electrical current. Thus, the narrow beam is a “hot” beam while the wide beam is a “cold” beam. The hot beam expands more than the cold beam, causing a lateral deflection in the free end.

According to the parameters given by ref. [23] (*t* = 2.0 μm, *w_h_* = 2.5 μm, *w_c_* = 16 μm, *l_f_* = 50 μm, *l_c_* = 180 μm, and *s* = 2.5 μm), we analyzed the deflection of the free end and the temperature distribution on the hot and cold arms using COMSOL Multiphysics [24]. The simulation results show good agreement with the experimental results from ref. [23], as shown in Figure 3b. The temperature distribution on the actuator given by COMSOL also agrees well with the results reported in ref. [25], as shown in Figure 3c.

Though the hot-cold arm has a relatively simple structure, it still needs an array for producing a large force to drive a mirror plate with a complicated design structure [26]. Moreover, a hot-cold arm actuator offers only a rotating motion but is difficult to generate a piston motion, which limits the application range. More hot-cold arm actuator designs will be presented in U Shaped Thermal Actuator-Based Micromirrors Section.

##### Chevron Actuators

Chevron actuators are well known as V-shaped actuators. The Chevron microstructure design was first introduced by Yogesh et al. [27] for strain sensors. Later, Cragun et al. [28] and Que et al. [29] adopted the V-shaped structure as electrothermal actuators. As shown in Figure 4, a typical Chevron actuator is made of a pair of slanted beams (i.e., a V shape) and a shuttle. The beams are typically made of polysilicon, thus forming a resistor. When a current passes through the beams, Joule heating is generated and consequently the temperature on the beams is increased, leading to thermal expansion of the beams. Due to the V-shaped configuration, a net force is generated along the symmetric axis by the beams. So, the shuttle moves longitudinally.

In order to increase the actuation force and overall robustness, multiple pairs of V-shaped beams are often employed, as shown in Figure 5a, where *L* and *w* are the length and width of the beams, respectively. α is the slanted angle of the V-shaped beams, and *d* is the distance between adjacent beams. Assuming the temperature change of the beams is Δ*T*, then the displacement, *d_e_*, of the shuttle is readily given by [30]:(1)de = (L2+2Lα1ΔT−L2cos2α)12 − Lsinα
where *α*_1_ is the thermal expansion coefficient (TEC) of the beam material. Figure 5b shows the temperature distribution of the V-shaped actuator under a DC voltage of 5 V. The structural parameters are shown in Figure 5a. It can be seen that the shuttle has an in-plane displacement of 1 μm. Note that the temperature distribution on the beams is not uniform with the highest temperature (~599 K) located near the middle of each beam. The displacement of the shuttle increases non-linearly with the increase of the drive voltage, as shown in Figure 5c. The temperature distribution of one V-shaped beam is shown in Figure 5d. The two beams have a symmetric temperature distribution and the highest temperature appears around the middle of each beam. More V-shaped actuator-based micromirrors designs will be presented in V-Shaped Thermal Actuator-Based Micromirrors Section.

#### 3.1.2. Bi-Material Solid Thermal Actuators

Thermal actuators can be constructed by stacking two materials with different thermal expansion coefficients (TECs). This is often called a bi-material or bimorph thermal actuator. The analytical model of a bimorph beam was proposed in the early 20th century by Timoshenko [31]. In 1988, Werner et al. introduced an Si-Au bimorph actuator integrated with a polysilicon heater and achieved a deflection of approximately 0.1 μm/K for potential applications in electric switches, microvalves and optical mirrors [32]. In 1989, Bencke et al. demonstrated a bimorph-based electrothermal micromirror [33]. Figure 6a shows a bimorph beam, where the thicknesses of the two materials are *t*_1_ and *t*_2_, respectively. When the temperature increases, the bimorph bends towards the material with lower TEC (Figure 6b). In Figure 6c, for a small angle, if the widths of the two materials are equal, the tip deflection, *d_f_*, of the bimorph is given by [34,35]:(2)$$\frac{1}{r}\;=\;\frac{6\left(t_{1}+t_{2}\right)\left(a_{2}-a_{1}\right)\Delta T}{\left(4t_{1}^{2}+4t_{2}^{2}+6t_{1}t_{2}+\frac{E_{1}t_{1}^{3}}{E_{2}t_{2}}+\frac{E_{2}t_{2}^{3}}{E_{1}t_{1}}\right)}\;(\alpha_{2}\;>\;\alpha_{1})$$
(3)df = r(1−cos(θT)) = r(1−cos(lbr))
where *l_b_* is the length of the bimorph, *a*_1_ and *a*_2_ are the TECs of material 1 and material 2, respectively. Δ*T* is the temperature change, *r* is the radius of curvature of the bimorph caused by Δ*T*, *θ_T_* is the angle of curvature, and *E*_1_ and *E*_2_ are the Young’s moduli of material 1 and material 2, respectively. If the two materials and Δ*T* are certain, *d_f_* is a function of the ratio of *t*_2_*/t*_1_. Figure 7 shows the tip deflections of a few thermal bimorph examples, where it is assumed that *t*_1_ = 1 μm and Δ*T* = 400 K. The material properties can be found in Table 1. The optimized thicknesses of layer 2, which are based on the maximum deflections of the corresponding bimorphs, have been marked on their curves in Figure 7.

In 2003, Jain and Xie reported a tip-tilt-piston (TTP) micromirror using two orthogonal pairs of bimorph actuators with a gimbal to decouple the two rotational axes [36]. Todd and Xie proposed an inverted-series-connected (ISC) bimorph actuator design and demonstrated a TTP micromirror without the need of a gimbal [37]. In 2007, Wu and Xie further demonstrated a gimbal-less TTP micromirror using a lateral shift free (LSF) electrothermal bimorph actuator design [38]. More details about micromirrors based on bimorph thermal actuator designs will be presented in Section 4.1.2.

### 3.2. Liquid-Based Thermal Actuators

For a liquid droplet on a solid surface, surface tension causes a deformation of the droplet when the temperature changes, which is a thermocapillary effect or Marangoni effect [39]. The droplet deformation changes the contact angle of the liquid droplet on the solid surface, which can be used to make an actuator for micromirrors.

Surface tension is a temperature-related parameter and its value can be given by a semiempirical relation [40]:(4)$$\gamma \;=\;\gamma_{0}\left(1-\frac{1}{T_{c}-T_{0}}\left(T-T_{0}\right)\right)$$
where *γ*_0_ is the surface tension at the initial temperature *T*_0_, and *T_c_* is the critical temperature under which the surface tension *γ* becomes zero. Assume there is a liquid droplet on a silicon substrate with two heaters respectively located on the two sides, as shown in Figure 8a. After injecting an electrical current into one of the heaters, the surface tension of the liquid droplet above this heater will decrease as the temperature increases due to the Joule heating, which will cause an inner motion from the heated side to the cool side, as illustrated in Figure 8b. The contact angle *θ_r_* is readily given by [40]:(5)$$\theta_{r}\;=\;arccos\frac{\gamma_{SG}-\gamma_{SL}}{\gamma_{LG}}$$
where *γ_SG_*, *γ_SL_*, and *γ_LG_* are the surface tensions at the solid-gas, solid-liquid and liquid-gas interfaces, respectively. The relationship between temperature *T* and the contact angle *θ* can be obtained by substituting Equation (4) into Equation (5). The temperature *T* is determined by the injected current or the applied voltage and the resistance of the heater as well. More details about micromirrors based on the thermo-capillary actuation will be given in Section 4.2.

### 3.3. Thermo-Pneumatic Actuation

One type of thermo-pneumatic actuators is composed of a cavity sealed by a flexible membrane. Figure 9a illustrates a thermo-pneumatic-based micromirror that is made of a silicon frame, a glass substrate, a heater, a polymer membrane, a gas cavity, and a mirror plate. The membrane seals the gas cavity and can be deformed by increasing the gas temperature using the heater. Then the mirror on the center of the membrane achieves a vertical displacement (Figure 9b), or realizes a tilt angle if the mirror is mounted on one side of the membrane (Figure 9c). More details about micromirrors based on the thermo-pneumatic actuation will be given in Section 4.3.

## 4. Electrothermal Micromirrors

All of the thermal actuation mechanisms discussed in Section 3 have been utilized to make MEMS micromirrors. According to the number of rotational axes, electrothermal micromirrors can be classified into phase-only (i.e., pure piston motion), 1-axis rotational, 2-axis rotational, and tip-tilt-piston (TTP, i.e., 2-axis rotational plus piston) micromirrors. In this section, MEMS micromirrors based on solid-state thermal actuators will be reviewed first, followed by liquid-state and gas-state thermal actuators consecutively. At the end, electrothermal MEMS micromirrors integrated with other types of actuation mechanisms such as electrostatic and electromagnetic actuators are also reviewed.

### 4.1. Solid-State Thermal Micromirrors

Most electrothermal MEMS micromirrors are solid-state. In the following, we will first introduce single-material solid-state thermal micromirrors, including U-shaped beam, V-shaped beam, buckle-beam, and three-beam based types. Then we will introduce bi-material solid-state thermal micromirrors.

#### 4.1.1. Single-Material Solid-State Thermal Micromirrors

##### U Shaped Thermal Actuator-Based Micromirrors

Hot-cold arm (usually a U shape) actuator-based micromirrors offer a way to manipulate optical beams and were first proposed in 1996 [41]. Typically, one U-shaped actuator can only generate limited driving force, so multiple U-shaped actuators are often grouped to form an actuator array to increase the driving capacity. For example, in 1996, Reid et al. reported a micromirror driven by a U-shaped actuator array [41], where the mirror plate was supported by a hinge and pulled by the U-shaped actuator array, where the mirror plate was able to rotate up to 15°. The Multi-User MEMS process [42] was used for the fabrication. Also in 1996 Reid et al. demonstrated another U-shaped actuator based micromirror [43], as shown in Figure 10a, where the U shape was vertically orientated, i.e., the hot arm was located on the top of the cold arm, causing the deflection in the vertical direction, as shown in Figure 10b. Still in 1996, Reid et al. reported a upright rotating micromirror [26] using a rotatory stepper motor system with a series of U-shaped actuators [44], as shown in Figure 11, respectively. This rotating micromirror, consisting of a U-shaped actuator array, a drive pawl, a yoke and a push pawl, was capable of realizing a full 360° rotation.

Although U-shaped thermal actuators provide a simple way for producing lateral deflection, extra mechanisms are needed to generate out-of-plane motion. So, some crucial specifications, such as fill factor and stability, have to be sacrificed to make angular scanning or in-plane rotating. This may be one of the main reasons why U-shaped thermal micromirrors are seldom researched in recent days.

##### V-Shaped Thermal Actuator-Based Micromirrors

Similar to U-shaped thermal micromirrors, Chevron (i.e., V-shaped) thermal micromirrors also need arrays of V-shaped beams to produce large force to push or pull their mirror plates. In 2003, Chen et al. proposed a V-beam-based optical switch that was composed of two pairs of V-beam actuators and a reflective shutter [45], as shown in Figure 12a,b [46]. The shutter could move bi-directionally with a maximum stroke of 36 μm at the voltage of 22.3 V. An SEM image of one V-beam is shown in Figure 12c [46]. In 2005, the same group reported another type of V-beam-based optical switch [46]. As illustrated in Figure 12d,e, the switch can operate in two states: transmission state and switching state. Figure 12f is an SEM image of a fabricated switch.

In order to induce angular motion, 1D and 2D V-beam electrothermal micromirrors with pre-bent torsion bars were proposed by Eun et al. in 2009 [47]. Both the 1D and 2D micromirror were fabricated from SOI wafers with 50 μm-thick device layers, as shown in Figure 13 and Figure 14, respectively. For the 1D micromirror, the mirror plate was supported by a pair of torsion bars that were connected with an array of V-shaped thermal actuators at the side near the mirror plate through a pair of 10 μm-thick silicon beams, as shown in Figure 13a. The thicknesses of the V-shaped actuators and torsion bars were 50 μm. Due to the thickness difference between the thin silicon beams and V-shaped actuators, the lateral force generated by the V-shaped thermal actuators was converted into a rotational motion via the torsional bars, as shown in Figure 13b. Figure 13c shows an SEM image of the 1D micromirror. A maximum optical rotational angle of 6.5° was achieved at only 13 V dc.

The 2D V-beam thermal micromirror was based on the same principle as that of the 1D micromirror described above. There were four V-shaped thermal actuators distributed around the mirror plate and connected with tortional bars through thin silicon beams, as shown in Figure 14a,b. Maximum optical scan angles of 5.4° and 5.2° were achieved at the x-axis and y-axis at 11 V dc, respectively.

This type of micromirrors is robust, but they typically suffer from complicated structures and low fill factor due to a large space occupied by the actuators and torsion bars and their response times are also relatively large (~20 ms).

##### Buckle-Beam Thermal Actuator-Based Micromirrors

Using a single solid material, buckle-beam-based micromirrors have also been studied and demonstrated. This type of micromirrors can be fabricated using simple surface micromachining processes and can achieve a large angular deflection under resonance. For example, in 2001, Sinclair et al. demonstrated buckle-beam-based 1D and 2D thermal micromirrors in which a series of buckled beams supported a torque beam that was connected with a mirror plate through a pair of arms [48,49]. When the buckle beams are heated by injecting a current, an out-of-plane displacement will be created due to the positive TEC of polysilicon (Figure 15a). The 1D mirror achieved a scan angle of 18° at the resonant frequency of 9 kHz (Figure 15b) while the 2D micromirror achieved a scan angle of over 20° at the resonant frequency of about 16 kHz (Figure 15c). However, the fill factor of this type of micromirrors is still low.

##### Three-Beam Thermal Actuator-Based Micromirrors

A three-beam thermal actuator is composed of three parallel beams. The outer two beams are named hot arms as they have higher temperature than the center beam. Since the two hot arms expand longer than the center cold beam, the actuator moves vertically at its free end. In 2007, Li et al. reported such a micromirror, as shown in Figure 16a, where an optical scanning angle of 10° was obtained at 18 V dc [50]. Another similar 2D micromirror reported by the same group is shown in Figure 16b [51,52], where four actuators were employed to support a gold-coated mirror plate (*Φ*2 mm). From Figure 16b, we can see that the fill factor of the mirror is significantly improved compared to the previous micromirrors. Moreover, the fabrication process is quite straightforward.

#### 4.1.2. Bi-Material Solid Thermal Micromirror

Bi-material actuators not only can provide a large force with low voltage (a few volts), but can also easily achieve out-of-plane motion with simple structures and fabrication processes. Thereby, bi-material thermal actuators are the preferred choice when making electrothermal micromirrors. Bi-material solid-state thermal actuators are typically composed of two layers with different materials whose thermal coefficients of expansion (TECs) are different. Bi-material beams in this case are often referred as bimorph beams. Thus, bi-material sold thermal actuators are also called bimorph actuators. Single bimorph actuators are common in the early versions of bi-material-based electrothermal micromirrors. Single bimorph beams have simple structures and can produce large scan angle, but they suffer from inevitable undesired large lateral shift [53]. Fortunately, the lateral shift is eventually overcome by more supplicated bimorph actuator designs [37,38].

##### Single Bimorph Thermal Micromirrors

In 1992, Buser et al. proposed IC-compatible Si/Al-based bimorphs as the actuator for making micromirrors [54], as shown in Figure 17a, where S1 and S5 could achieve 2D scanning while S2 and S3 had 1D scanning ability and S4 and S6 were some basic testing structures. However, the fabrication processes were difficult to obtain a thick Al layer on a thin silicon beam. Standard CMOS processes were adopted to fabricate the bimorph-based micromirrors. In 1995, Buhler et al. reported a 1D electrothermal micromirror based on SiO_2_/Al bimorphs with a polysilicon heater embedded [55], as shown in Figure 17b, where the mirror achieved a deflection angle of 0.3° at the power of 6 mW and the response time was 2 ms. Later, the same micromirror design was optimized and achieved a deflection angle of approximately 1° [56]. In 1999, to improve the scanning angle, Schweizer et al. optimized the thickness ratio of the bimorph layers and found that the optimal Al/SiO_2_ thickness ratio was 0.4 [53], as shown in Figure 17c; some of the fabricated mirrors are shown in in Figure 17d,e, where their resonant frequencies were in the range from 100 to 600 Hz and their maximum mechanical deflection angles reached above 90°.

Note that the mirror plates of the bimorph-based micromirrors discussed above are all made of thin films whose residual stresses seriously limit the maximum size of the mirror plates (typically less than 300 μm). Also the mirror plates are typically supported by only two bimorph beams. In order to increase the mirror plate size without sacrificing the robustness of bimorph-based micromirrors, Xie et al. proposed and demonstrated an 1D electrothermal micromirror with an array of SiO_2_/Al bimorphs and a thick mirror plate supported by a single-crystal silicon layer [57], as shown in Figure 18a. An SEM of such a fabricated micromirror is shown in Figure 18b, where the mirror plate was as large as 1 mm because of the 40 μm-thick silicon layer and was robustly supported by a mesh of bimorph beams. The resonant frequency of the mirror was 165 Hz, and the optical scanning angle reached above 25° at 10 mA dc.

However, the micromirror described above had large initial tilt angle, making its packaging difficult. Also, this micromirror’s rotation was unidirectional. In 2004, Jain et al. reported a bi-directional electrothermal micromirror with a pair of opposing bimorph arrays [58]. The mirror plate stayed flat at the rest position, as shown in Figure 18c,d. In addition to bi-directional scanning capability, this micromirror was able to generate large vertical displacement; under a drive voltage of 6 Vdc, the vertical displacement reached 200 μm. Due to its flat mirror plate and vertical and bi-directional scanning ability, the micromirror in ref. [58] is suitable for endoscopic optical probes and miniature interferometry systems [59].

Although 1D micromirrors have simple fabrication processes, they are not applicable in 3D imaging applications in which bi-axial (2D) scanning is needed. To realize 2D scanning, multiple bimorph actuators are usually required. In 2003 and 2004, Jain et al. devised a 2D micromirror that consisted of two actuators with a gimble [36,60]. One of the actuators is connected with a frame and the substrate, whereas the other actuator is attached between the mirror plate and the frame, as shown in Figure 19. The thickness of the mirror plate is approximately 45 μm, which results in a high flatness and robustness. The released mirror realized the scanning angles of 45° and 25° about the first-axis and second-axis at 15 V and 17 V, respectively. It should be noted that this 2D micromirror still has large initial tilt angles.

In order to decrease the unwanted initial tilt angle, some electrothermal micromirrors with spring connectors were proposed. For example, in 2005, Singh et al. proposed a single crystal silicon bimorph-based electrothermal micromirror [61], as shown in Figure 20a,b, where four thermal actuators (Si/Al bimorphs) were connected to a central mirror plate via four flexure springs. The thermal response time of the micromirror was 8–13 ms and the deflection angle reached 10° at an excitation voltage of below 1.5 V. Later, the same group improved the design and more micromirrors were reported in refs. [62,63,64,65], some of them are shown in Figure 20c–e. Those types of micromirrors rely on the flexure springs to absorb the lateral shift and tilt of the bimorph beams but at the price of scanning stability and robustness.

##### Triple Bimorphs Thermal Micromirrors

Most of aforementioned micromirrors can realize large displacements or scanning angles. However, they also generate undesirable effects such as large initial tilt angle and/or lateral shift which can cause optical path shift and the light beam may miss the mirror plate, as illustrated in Figure 21a.

To eliminate the lateral shift and initial tilt, in 2007, Wu et al. proposed a type of electrothermal micromirror with three segments of bimorphs [38,66]. The schematic diagram of the micromirror, consisting of three Al/SiO_2_ actuators, two frames and a mirror plate, is shown in Figure 21b. The schematic of the geometric structure is illustrated in Figure 21c, and the lateral shift of the mirror plate can be canceled when the length of actuator 3 is 2 times as large as that of actuator 1 or actuator 2, and the length of frame 1 and frame 2 are equal. For a released micromirror, a piston motion of 0.62 mm was achieved at 5.3 V dc, whereas the lateral shift was only 10 μm, as shown in Figure 21d. In 2011, Liu et al. fabricated a micromirror with a very small device footprint of 1.5 × 1.5 mm^2^ whose scanning angle reached ±16° at a low voltage of less than 3.6 V dc [67], as shown in Figure 21e. In 2012, the same group reported a mirror which reached optical angles of at the driving voltage of 4.5 V [68]. The bimorphs of these two types of micromirrors had the same materials of Al/SiO_2_.

Cu/W bimorphs are also used in electrothermal micromirrors. For example, in 2015, Zhang et al. proposed a Cu/W electrothermal micromirror, which generated a large vertical displacement of 320 μm and a maximum optical tilt angle of ±18° at 3 V, as shown in Figure 21f [69]. Moreover, the response time was 15.4 ms that was lower than the response time (25 ms) in ref. [66].

All the above micromirrors in this subsection have rectangular mirror plates. However, in fact, the spots from most of light sources are circular or elliptical, which means that micromirrors with circular mirror plates are more preferable than the rectangular mirror plates to reflect light beams. In 2011, Liu et al. devised and demonstrated two types of circular mirrors with a pair of curved concentric bimorph actuators and circular mirror plates [70,71], as shown in Figure 22a,b. For the two types of micromirrors, each circular mirror plate was supported by Al/W-based curved concentric actuators. Type-I (Figure 22a) achieved the scanning angles of up to ±11° at 0.6 V and piston motion of 227 μm with only 7 μm lateral shift at 0.8 V, respectively. For Type-II (Figure 22b), the piston-only motion of 200 μm was achieved at 0.9 V, and the corresponding lateral shift was less than 3 μm.

##### Quadruple Bimorphs Thermal Micromirrors

The aforementioned triple bimorph micromirrors overcome the lateral shift issue effectively, but there still exists in-plane rotation of the mirror plate during piston scanning. Benefited from some previous reports from references [72,73], in 2005, Todd et al. demonstrated an inverted-series-connected (ISC) electrothermal micromirror that eliminated lateral shift and in-plane rotation [37]. Each ISC actuator is composed of two S-shaped bimorphs, each of which consists of two segments of inverted bimorphs, as shown in Figure 23a. Elevating the temperature on the actuator, the lateral shifts caused by S1 and S2 will cancel each other and B point only moves in the *z*-axis. For a released micromirror, the mirror plate can be displaced by 70 μm at the approximate temperature of 130–140 °C, as shown in Figure 23b.

However, the displacement of the mirror in [37] is limited by the metal sidewall pileup on the S-shaped actuator. In 2009, Jia et al. improved the design by optimizing the thickness ratio of the bimorph, the length ratio and the overlap ratio of the S-shaped actuator, and fabricated a micromirror with a piston motion of 480 μm and an optical tip-tilt angle of ±30° at less than 8 V, as shown in Figure 24a [74]. To obtain a micromirror with large mirror plate, in 2015, Wang et al. proposed a meshed ISC micromirror with a downward mirror plate (size: 1.06 mm × 1.06 mm) (Figure 24c) that could generate a piston motion of 145 μm at 1.8 V dc and had a response time of just 4 ms [75]. The large mirror plate can provide an effective aperture to reflect light beams. In order to further increase the robustness, in 2015, Xie et al. proposed a micromirror with multilevel actuators [76], and the micromirror (Figure 24b) consisted of sixteen three-level ladder actuators and a 1.4 mm × 1.2 mm of mirror plate, which generated a piston motion of 300 μm at 10 Hz and drive voltage of 7 Vpp. In 2018, according to the report from Wang et al., another micromirror (Figure 24d) with a large mirror plate (*Φ* = 2 mm) had the capability of achieving optical scanning angles of ±2.8° [77]. In 2019, another large micromirror (Figure 24e) with a 2 mm × 2.5 mm mirror plate could generate optical angles of 7° about both axes at 9 V dc [78]. Notably, in 2019, Wang et al. also devised and demonstrated an omnidirectional scanner with a tripod micromirror that achieved a mechanical tip-tilt angle of 16° [79], as shown in Figure 24f. The bimorphs of all of above micromirrors were made of SiO_2_/Al.

Although the SiO_2_/Al bimorph provides a large difference in the coefficient of thermal expansion, the melting point of Al is just at 660 °C, which limits the maximum operation temperature, and the bimorph needs an embedded polysilicon or Pt as its heater, which leads to the inability to provide a symmetric bimorph configuration. To solve these issues, in 2015, Zhang et al. introduced a Cu/W-based electrothermal micromirror (Figure 25a) with ISC actuators [80]. The mirror can not only generate a relatively large piston motion of 169 μm at 2.3 V dc, but can also improve its thermal response time (7.5 ms), and the resonant frequencies are 1.48 kHz for piston motion and 2.74 kHz for tip-tilt motion. In 2017, Wang et al. improved the next generation of this type of micromirror (Figure 25b) in terms of its resonant frequencies [81]. The resonant frequencies reach at 7.6 kHz for piston motion and 12.8 kHz for tip-tilt motion. However, this type of micromirror still has some issues. For example, in 2017, as depicted by Tanguy et al., copper is easy to oxidize over time [82], as shown in Figure 25c,d.

Different types of electrothermal bimorphs can also be combined for the micromirrors. For example, Quentin et al. introduced a 2-axis MEMS micro scanner with large scanning ranges of the frame (32°) and the mirror (22°) using ISC quadrable bimorphs and single bimorph arrays to tip and tilt a dual-reflective mirror plate [83,84], as shown in Figure 26.

##### An Origami Inspired Bimorph Thermal Micromirror

In 2021, Masaaki et al. reported an electrothermal scanner, inspired by traditional Japanese origami, consisting of eight curved NiCr/SiN-bimorph actuators and a mirror plate (*Φ* = 2 mm) with a hole [85], as shown in Figure 27a. The released scanner (Figure 27b) generated a piston motion of 200 μm at a voltage of 25 V dc or at a power of 131 mW, as shown in Figure 27c. Note that this origami inspired micromirror can only move vertically.

Bi-material-based micromirrors play the main role among various types of electrothermal micromirrors. The LSF-based and ISC-based actuators are proposed to improve the reliability by suppressing the lateral shift, both of which are the most commonly-reported micromirrors in recent days. Large scan range at low voltage is the clear feature of bi-material-based electrothermal micromirrors. However, repeatability and stability are the bottlenecks for electrothermal micromirrors. Proper materials should be adopted to improve the performance of bi-material-based actuators.

### 4.2. Liquid-Based Thermal Micromirrors

Compared with solid-state micromirrors, liquid-based thermal micromirrors are easy to fabricate without additional complicated MEMS processes. In 2009, in the report from Dhull et al., a liquid microdroplet was dropped on a Teflon surface which adhered to a silicon chip; the droplet could generate a motion by heating one of its sides [17], as shown in Figure 28a. Due to the thermocapillary effect, a mirror plate was tilted by the droplet, achieving a maximum angle of 6.5° at 30 V with a frequency of 7 Hz, as shown in Figure 28b,c.

Notably, the liquid droplet-based micromirrors are limited in robustness since the liquid is vulnerable to ambient vibrations. Moreover, many types of liquid droplets may evaporate quickly under high temperature. Thus, liquid with high boiling point is necessary for this type of micromirrors [86].

### 4.3. Thrmo-Pneumatic Micromirrors

The advent of thermal-pneumatic micromirrors has enriched the micromirrors families. In 2006 and 2008, Weber and Zappe reported tilting and piston-scanning micromirrors based on thermo-pneumatic actuation [18,87], as shown in Figure 29a–c, where the scanning angle was as large as 12.5° at a voltage of 30 V and the maximum vertical displacement was about 80 μm at 20 V. The same group also demonstrated another thermo-pneumatic micromirror with a large mirror plate of 5 mm in diameter [88], as shown in Figure 29d,e, where a tip-tilt deflection of 5° and a piston motion of 385 μm were achieved under a drive voltage of 35 V. However, these thermo-pneumatic micromirrors have several issues. For example, the reflectivity of their mirror plates was low because a layer of PDMS was coated on the mirror surface. Another issue is their slow response.

### 4.4. Electrothermal Actuator-Base Hybrid Micromirrors

Electrothermal actuators can provide large scanning angle and/or large vertical displacement, but they have high-power consumption and slow response. In contrast, electrostatic actuators perform quite well in terms of power consumption and response time; electromagnetic actuators can have large scanning angles and can be operated at high speed. Combining electrothermal actuators with electrostatic or electromagnetic actuators may overcome their limitations.

In 2011, Li et al. adopted three-beam thermal and electrostatic actuators to form a hybrid micromirror which was fabricated on an SOI wafer with a 10 μm-thick device layer. The micromirror generated a scanning angle of 10° at 15 V by its thermal actuators and a 24° scanning angle at the resonant frequency of 1.58 kHz by its inner electrostatic actuators [89], as shown in Figure 30a. The response of the electrostatic actuators was approximately 50 times faster than that of the electrothermal actuators.

Zhang et al. reported two electrothermal-electrostatic hybrid micromirrors in 2013 and 2016, respectively, both of which employed electrothermal actuators for the out-of-plane and electrostatic actuators for the in-plane motion [21,90]. Figure 30b shows the first generation of hybrid micromirror, which realized a vertical displacement of 370 μm at only 2.5 V but with very small in-plane motion. For the second generation of hybrid micromirror, as shown in Figure 30c, the in-plane displacement was over 40 μm at 109.6 Vdc and the out-of-plane displacement was 100 μm at 13.5 Vdc.

Another hybrid scanner composed of electrothermal and electromagnetic actuators was proposed by Kah et al. in 2012 [20], as shown in Figure 31a,b. The electrothermal actuator was made from an SOI wafer’s device layer to form a silicon cantilever beam and an aluminum heater was deposited on the cantilever beam. The electrothermal actuator was employed in a slow scanning axis, which achieved an optical deflection angle of ±1.5° at 12 mW at 74 Hz. The electromagnetic actuation was formed by a pair of permanent magnets and a multi-turn aluminum coil that was embedded in the frame, which achieved the scanning angles of ±10° at 1 Vpp with a frequency of 202 Hz.

Additionally, Chevron actuators have been utilized to adjust the resonant frequency in a hybrid micromirror. In 2011, Eun et al. reported a tunable micromirror [91], as shown in Figure 32a,b. An angular vertical comb-drive (AVC) actuator with a permanent initial tilt angle between its rotor and stator combs can generate a larger torsional displacement than the staggered vertical comb-drive (SVC) actuator with similar combs. An electrothermal actuator was used to form the permanent tilt angle in the electrostatic combs, which reached a maximum optical tilt angle of 30.7° with a 4.56 kHz sinusoidal voltage of 0–80 V. In 2015, in a work reported by Lee et al., Chevron electrothermal actuators were integrated on the torsional bars of an electrostatic comb actuated micromirror [92]. The thermal actuator was used to change the torsional stiffness by stressing the hinges so that the micromirror could operate at a variable resonant frequency by a range of 3%, as shown in Figure 32c,d.

### 4.5. Summary

The development of different types of thermally-actuated micromirrors is summarized in the timeline in Figure 33. Bimorph-based micromirrors first emerged at the end of 1980’s. Subsequently U-shape-based micromirrors with complicated mechanical structures were studied for optical switching applications. Soon after that, bimorph-based micromirrors were refocused on bimorphs with innovative structure designs such as LVD, LSF and ISC. At the same time, thermo-pneumatic and thermocapillary micromirrors were proposed. Also, electrothermal actuators combined with electrostatic or electromagnetic actuators emerged in early 2010’s. Overall, bimorph-based thermal micromirrors are dominant in the course of development of electrothermal micromirrors. Different types of electrothermal micromirrors are further summarized in Table 2.

## 5. Conclusions and Future Perspectives

Electrothermal actuators are devices that can transform electric energy to thermal energy and then to mechanical energy. These MEMS devices are promising in myriads of applications, such as micro-tweezers, micro-needles and micromirrors. This article provides a comprehensive review of different types of electrothermal micromirrors based on U-shaped actuators, V-shaped actuators, buckle-beam actuators, three-beam actuators, bimorph-based actuators, liquid droplet-based actuators, and thermo-pneumatic actuators. In general, U-shaped and V-shaped actuator-based micromirrors have complicated structures as they must use multiple joints to transfer in-plane motion into out-of-plane motion, and these micromirrors typically can only achieve very limited angular scan range. Buckle-beam thermal actuator-based micromirrors can be made by simple surface micromachining processes, but those micromirrors have several disadvantages such as low fill factor and low robustness. Three-beam-based micromirrors can achieve versatile motions, but their scan range need to be improved for practical use. Bi-material-based micromirrors receive the most attention due to their simple fabrication and large scan range. Liquid droplet-based and thermo-pneumatic micromirrors provide alternative options but their robustness is a major concern and they are sensitive to ambient vibrations and disturbances as well.

Among all these types of electrothermal micromirrors, bimorph-based micromirrors exhibit clear advantages, such as large scan range, mechanical robustness, and simple fabrication process. Their undesirable effects, such as large lateral shift and initial tilt, accompanied by single biomorphs, can be compensated by multi-bimorph designs, such as LSF actuators and double-S bimorphs actuators. Moreover, different materials are also investigated for bimorph actuators to improve their responsivity and reliability. Although SiO_2_ and Al bimorphs exhibit a large difference in the coefficient of thermal expansion, this combination is limited in applications due to the low melting temperature of aluminum. In comparison, Cu/W bimorphs can operate at much higher temperature.

Moreover, electrothermal actuator-based hybrid micromirrors are also discussed in this review. It should be noted that ideal micromirrors should have the merits of high speed, large scan range, low drive voltage, high fill factor, large optical aperture, and multiple degrees of freedom. The high speed of electrostatic actuation and electromagnetic actuation has been exploited to make up the relative long thermal response time of electrothermal actuation. In addition, the repeatability and stability are also the important parameters that researchers must carefully consider when developing as thermal micromirrors.

## Figures and Tables

**Figure 1 micromachines-13-00429-f001:**
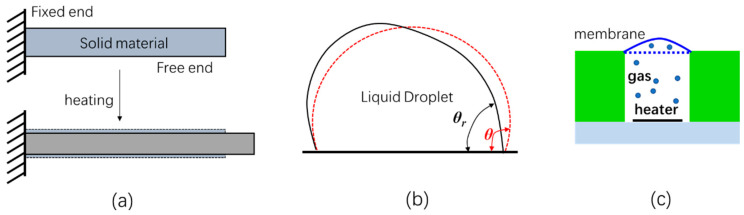
Schematical diagrams of (**a**) solid-based actuation, (**b**) liquid droplet-based actuation and (**c**) thermal-pneumatic actuation.

**Figure 2 micromachines-13-00429-f002:**
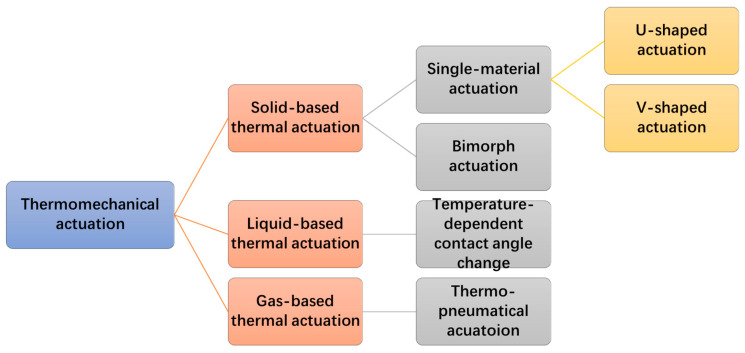
The classification diagram of thermomechanical actuations.

**Figure 3 micromachines-13-00429-f003:**
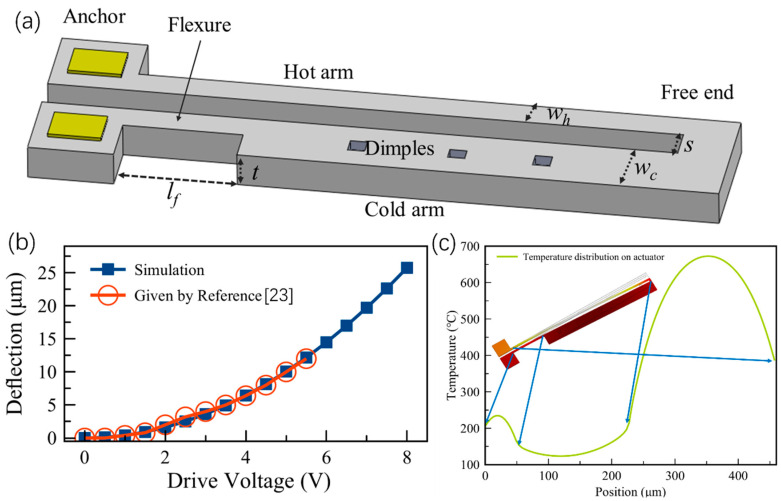
(**a**) Schematic diagram of a hot-cold arm actuator; (**b**) the simulation and theoretical [23] results of the actuator’s deflections; (**c**) temperature distribution on the hot-cold arm actuator.

**Figure 4 micromachines-13-00429-f004:**
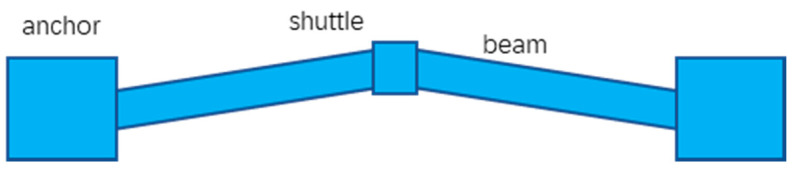
The schematic diagram of a V-shaped actuator.

**Figure 5 micromachines-13-00429-f005:**
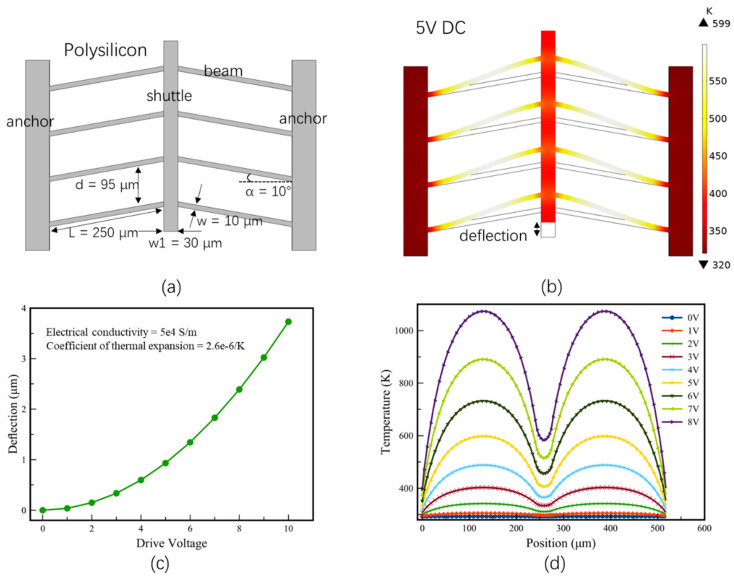
(**a**) Basic design of a Chevron actuator; (**b**) simulation results of the temperature distribution and shuttle deflection under drive voltage of 5 V; (**c**) deflection vs. drive voltages from COMSOL modeling; (**d**) temperature distribution on two consecutive beams and shuttle vs. their positions under different drive voltages.

**Figure 6 micromachines-13-00429-f006:**
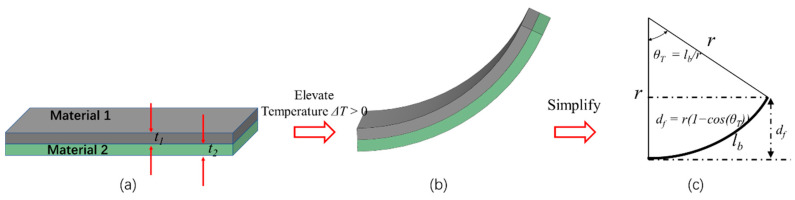
(**a**) Schematic diagram of a bimorph; (**b**) the deflection in the tip after elevating temperature; (**c**) curvature calculation of a curled beam.

**Figure 7 micromachines-13-00429-f007:**
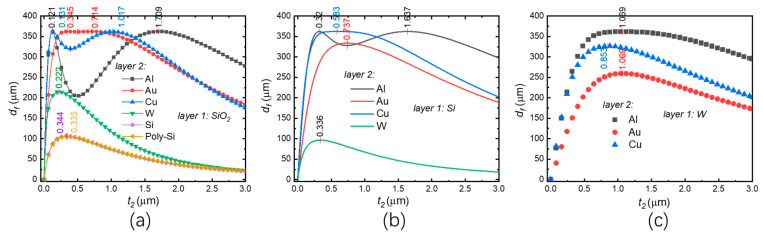
Tip deflection vs. the thickness of layer 2 for (**a**) the SiO_2_-based bimorphs; (**b**) the Si-based bimorphs and (**c**) the W-based bimorphs.

**Figure 8 micromachines-13-00429-f008:**
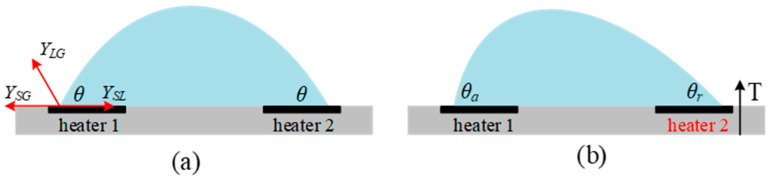
(**a**) A liquid droplet on a silicon substrate with heaters under the droplet; (**b**) contact angles are changed by injecting a current into the heater 2.

**Figure 9 micromachines-13-00429-f009:**
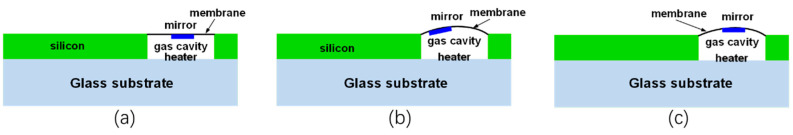
(**a**) A micromirror with thermo-pneumatic actuator; (**b**) the tilt-tip motion of the mirror plate; (**c**) the piston motion of the mirror plate.

**Figure 10 micromachines-13-00429-f010:**
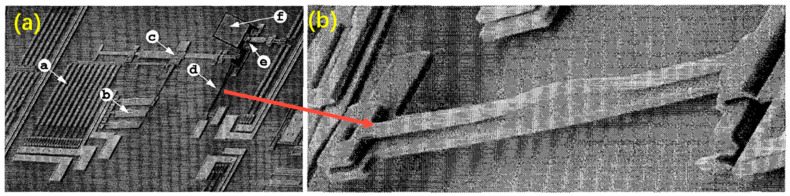
(**a**) A micromirror with a vertical actuator; (**b**) a closed-up view of the vertical actuator [43]. (Reprinted with permission from IEEE).

**Figure 11 micromachines-13-00429-f011:**
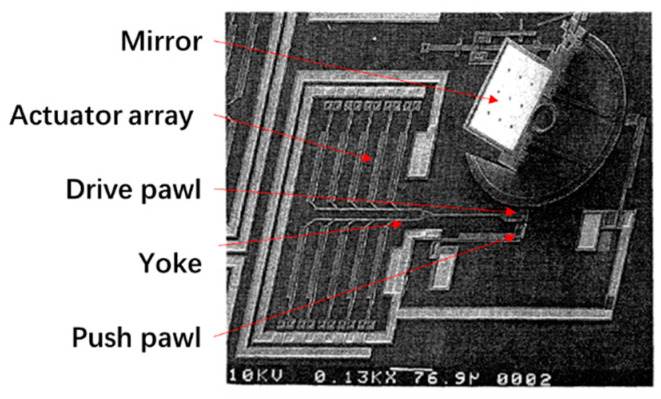
Rotating micromirror (Reprinted from Reid et al. [26] with permission from IEEE).

**Figure 12 micromachines-13-00429-f012:**
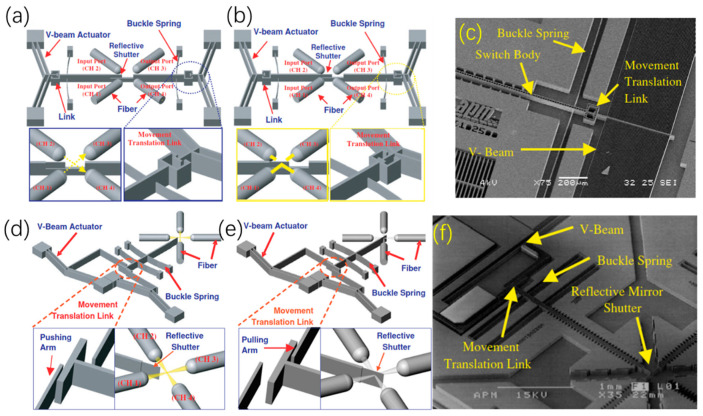
Schematic diagram of the optical switch for (**a**) the transmission state and (**b**) the switching state; (**c**) SEM photos of the optical switch. Schematic diagram of another optical switch for (**d**) the transmission state and (**e**) the switching state; (**f**) SEM photos of another optical switch. (Reprinted from Lee C et al. [46] with permission of IOP).

**Figure 13 micromachines-13-00429-f013:**
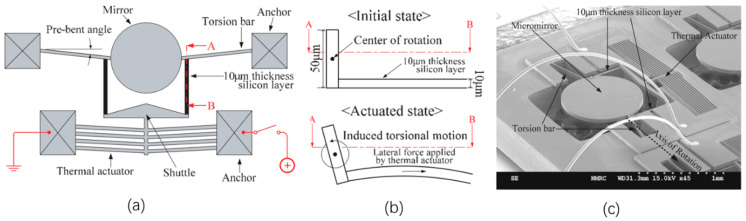
(**a**) Schematic diagram of a 1D micromirror; (**b**) the mechanism for converting the lateral motion to torsional motion; (**c**) SEM image of the 1D V-beam electrothermal micromirror. (Reprinted from Eun et al. [47] with permission of IOP).

**Figure 14 micromachines-13-00429-f014:**
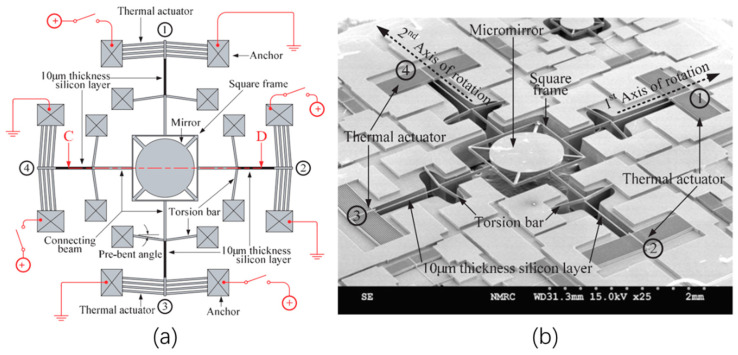
(**a**) Schematic diagram of the 2D V-beam thermal micromirror; (**b**) SEM image of the 2D V-beam electrothermal micromirror. (Reprinted from Eun et al. [47] with permission of IOP).

**Figure 15 micromachines-13-00429-f015:**
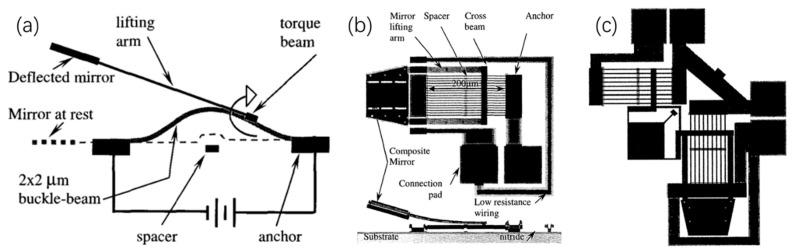
(**a**) Schematic diagram of a buckle-beam actuated micromirror; (**b**) 1D buckle-beam-based micromirror; (**c**) 2D buckle-beam-based micromirror. (Reprinted from Sinclair et al. [49] with permission from IEEE).

**Figure 16 micromachines-13-00429-f016:**
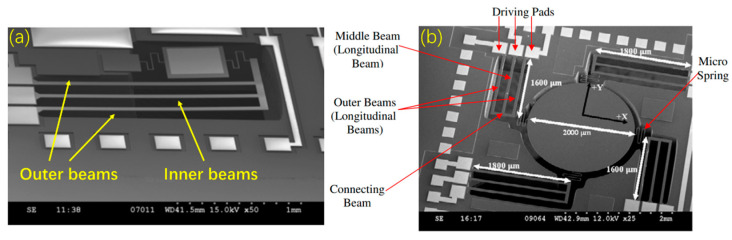
SEM photos of (**a**) a 1D micromirror (Reprinted from Li et al. [50] with permission of IOP) and (**b**) a 2D micromirror. (Reprinted from Li et al. [52] with permission of IOP).

**Figure 17 micromachines-13-00429-f017:**
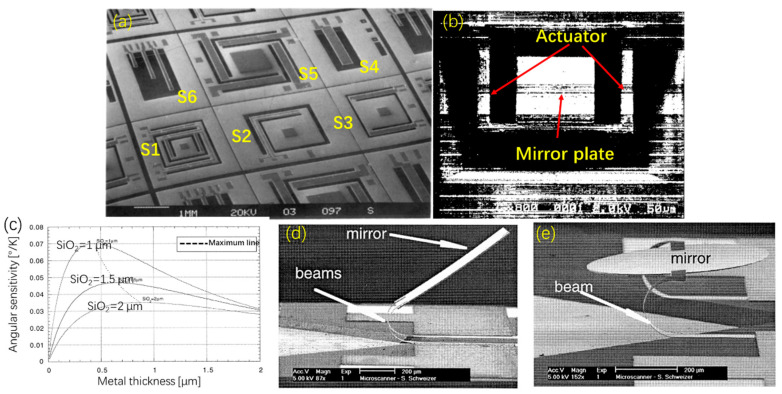
(**a**) SEM photo of multiple types of thermal micromirrors [54]; (Reprinted with permission of Elsevier) (**b**) SEM photo of micromirror (Reprinted with permission of Elsevier) and (**c**) actuator sensitivity versus metal thickness and (**d**) SEM photo of mirror with 500 μm by 800 μm and (**e**) SEM photo of the lateral fixed beams [53] (Reprinted with permission of Elsevier).

**Figure 18 micromachines-13-00429-f018:**
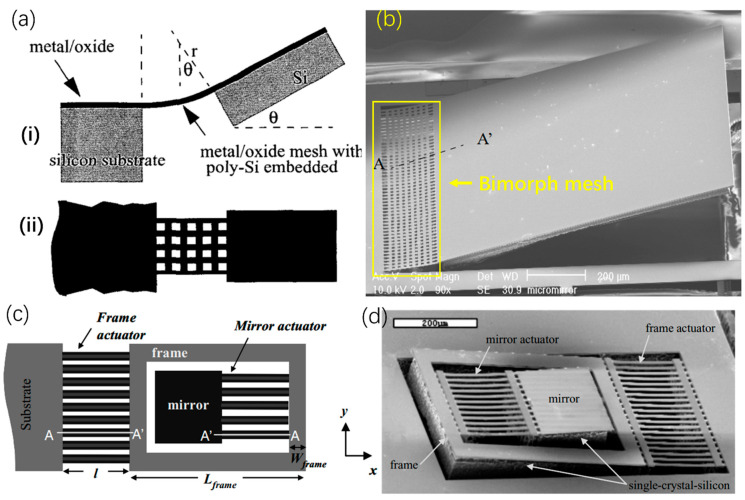
(**a**) The conceptual design of 1D micromirror with thick mirror plate (**i**) cutaway view and (**ii**) top view; (Reprinted with permission from IEEE) (**b**) SEM photo of micromirror after realizing (Reprinted with permission from IEEE) and (**c**) schematic diagram of the vertical displacement micromirror and (**d**) SEM photo of the mirror device [58] (Reprinted with permission from IEEE).

**Figure 19 micromachines-13-00429-f019:**
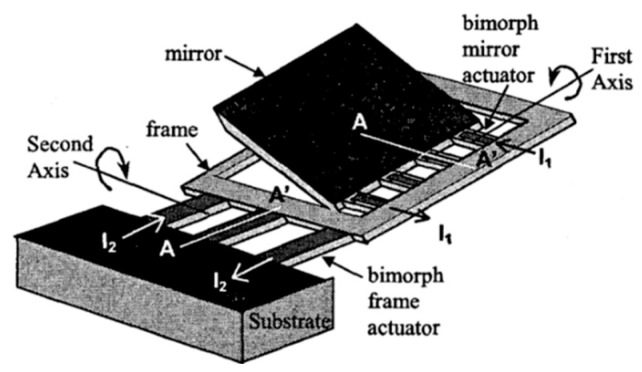
Schematic diagram of the 2D mirror [36]. (Reprinted from Jain et al. [36] with permission from IEEE).

**Figure 20 micromachines-13-00429-f020:**
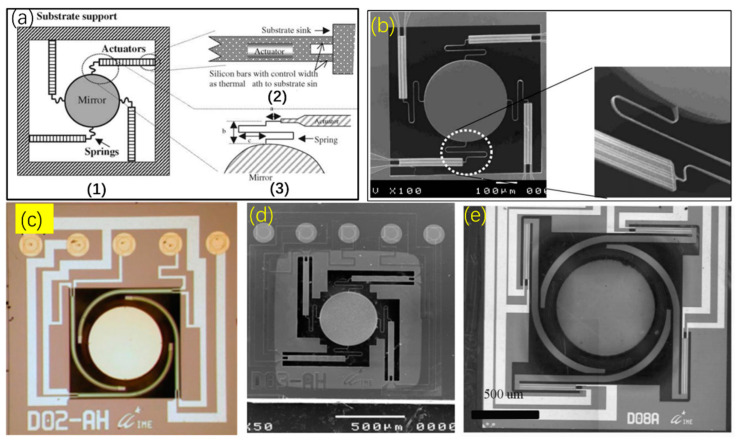
(**a**) Schematic diagram of Si/Al bimorph micromirror and its close-up view; (**b**) SEM image of the micromirror with four actuators and springs and its close-up view of the spring connecter; (Reprinted with permission from Singh et al. [61] of Elsevier) (**c**) optical image of a released 3D micromirror chip [63] (Reprinted with permission from IOP); (**d**) SEM image of micromirror with linear actuator [64] (Reprinted with permission from IOP); (**e**) micromirror with folded actuators [65] (Reprinted with permission from IEEE).

**Figure 21 micromachines-13-00429-f021:**
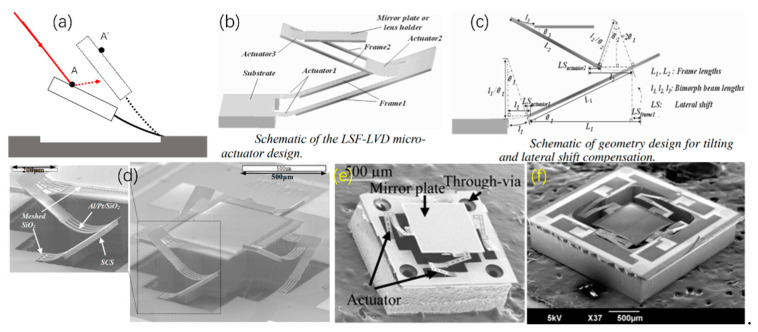
(**a**) A lateral shift would cause an optical beam missing the mirror surface; (**b**) Schematic of the LSF-LVD micromirror; (**c**) a simplified diagram (Reprinted with permission from IEEE); (**d**) SEMs of the LSF micromirror and the close-up views of the LSF actuator (Reprinted with permission from IEEE); (**e**) SEM of the electrothermal micromirror with TSVs (Reprinted with permission from Liu et al. [67]); (**f**) the micromirror with Cu/W bimorphs [69].

**Figure 22 micromachines-13-00429-f022:**
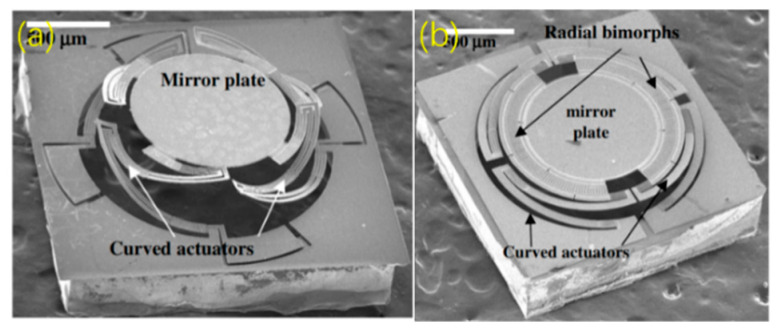
SEMs of the micromirrors with (**a**) tip-tilt-piston motion [70] (Reprinted with permission from IEEE) and (**b**) piston-only [71] (Reprinted with permission from IEEE).

**Figure 23 micromachines-13-00429-f023:**
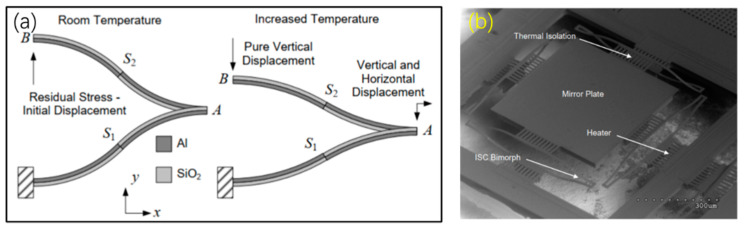
(**a**) Schematic diagram of an S-shaped bimorph actuator; (**b**) SEM image of a released S-shaped micromirror [23] (Reprinted with permission from IEEE).

**Figure 24 micromachines-13-00429-f024:**
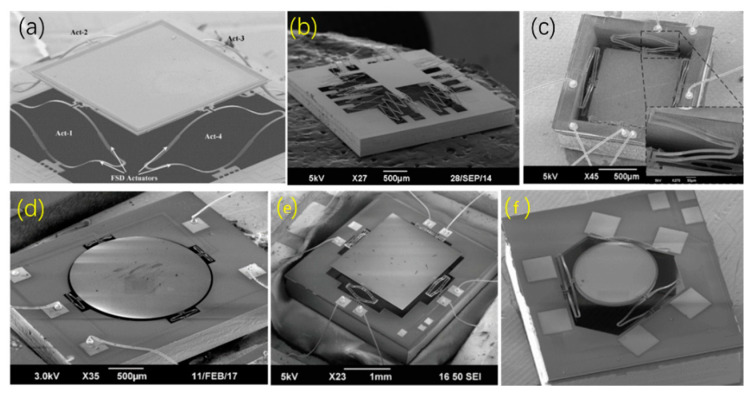
SEM images of the SiO_2_/Al-based micromirrors (**a**) with a 40 μm-thick silicon support (Reprinted with permission from IEEE); (**b**) a three-level ladder actuator (Reprinted with permission from IEEE); (**c**) a mesh ISC actuator (Reprinted with permission from IEEE); (**d**) with a large mirror plate; (**e**) with a 2 × 2.5 mm^2^ mirror plate (Reprinted with permission from IEEE); and (**f**) with tripod actuators [79] (Reprinted with permission from IEEE).

**Figure 25 micromachines-13-00429-f025:**
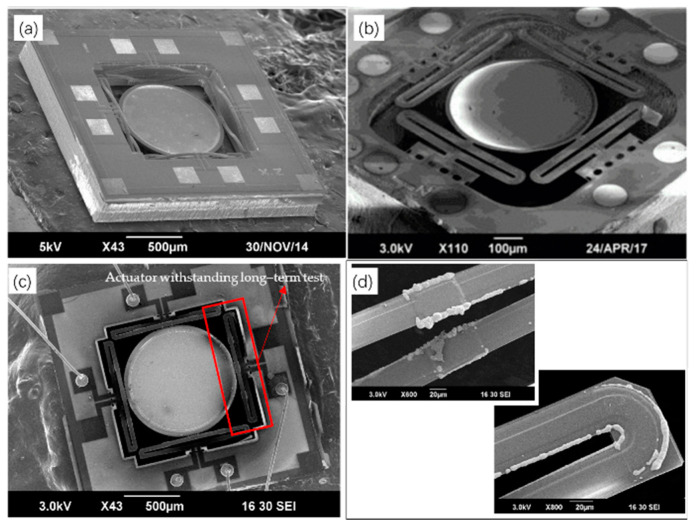
SEM images of Cu/W-based micromirrors (**a**) from ref. [80] (Reprinted with permission from IEEE); (**b**) from ref. [81] (Reprinted with permission from IEEE); (**c**,**d**) from ref. [82].

**Figure 26 micromachines-13-00429-f026:**
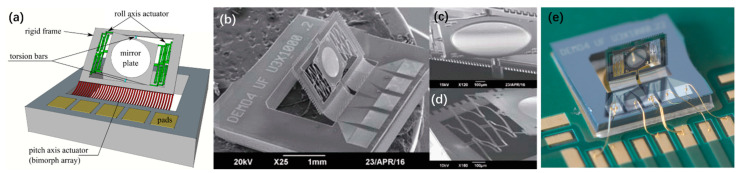
(**a**) Schematic design of the 2-axis micromirror with a torsion beam [84] and (**b**) its SEM image and (**c**,**d**) the close-up view of the backside mirror plate and meshed actuator; (Reprinted from Tanguy et al. [83] with permission from IEEE) (**e**) optical picture of a packaged micromirror device [84].

**Figure 27 micromachines-13-00429-f027:**
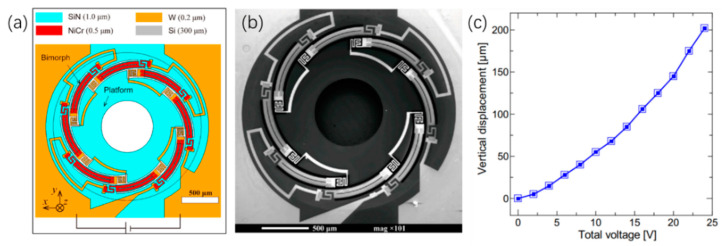
(**a**) Backside view of the mirror inspired by Japanese origami; (**b**) SEM images of the scanner; (**c**) curve of vertical displacement versus total voltage [85].

**Figure 28 micromachines-13-00429-f028:**
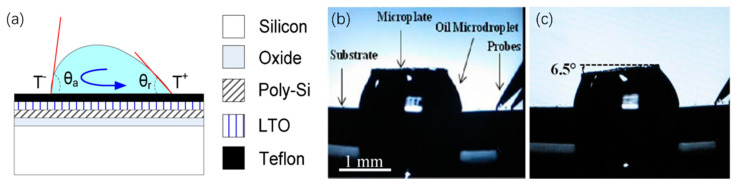
(**a**) The schematic diagram of the thermocapillary actuation; (**b**) microplate with drive voltage of 0 V and (**c**) with drive voltage of 30 V reaching a tilted angle of 6.5°. (Reprinted from Dhull et al. [17] with permission from IEEE).

**Figure 29 micromachines-13-00429-f029:**
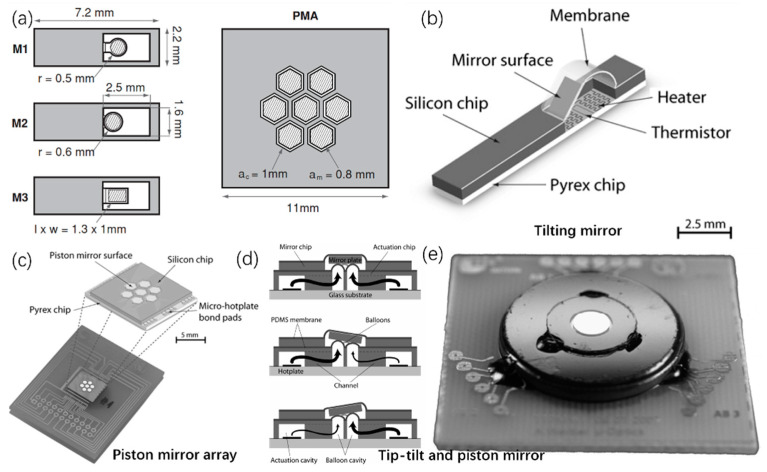
(**a**) Top view of the tilting mirrors and piston mirror array; (**b**) 3D sketches of a tilting mirror; (**c**) a piston mirror array; (Reprinted from Armin et al. [18] with permission from IOP) (**d**) schematic of a tip-tilt-piston micromirror; (**e**) a micrograph of the tip-tilt-piston micromirror. (Reprinted from Werber et al. [88] with permission from IEEE).

**Figure 30 micromachines-13-00429-f030:**
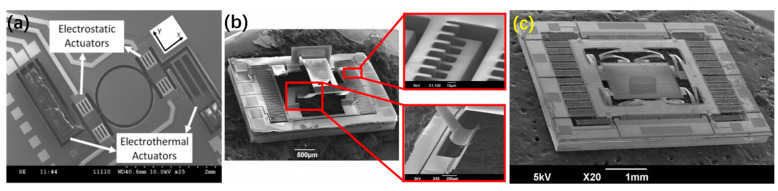
SEM images of hybrid micromirror with electrothermal and electrostatic actuators (**a**) from the report of Li et al. [89] (Reprinted with permission from IEEE) (**b**) from the report of Zhang et al. [21] (Reprinted with permission from IEEE) (**c**) from the report of Zhang et al. [90]. (Reprinted with permission from IEEE).

**Figure 31 micromachines-13-00429-f031:**
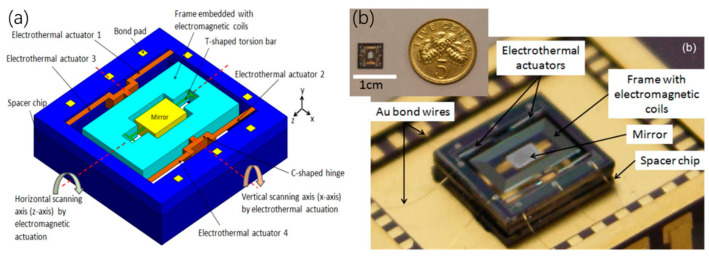
(**a**) 3D model of a hybrid micromirror with electrothermal and electromagnetic actuators and (**b**) its optical photo. (Reprinted from Kah et al. [20] with permission from IEEE).

**Figure 32 micromachines-13-00429-f032:**
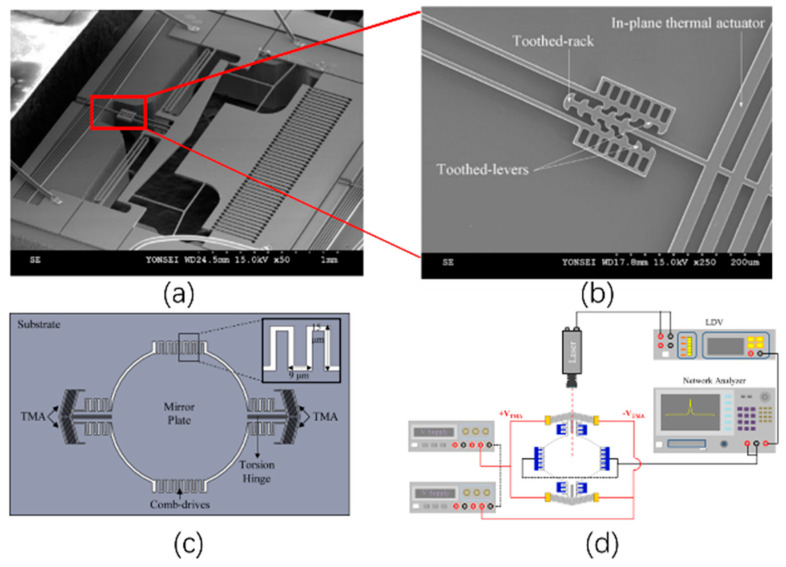
(**a**) SEM image of the hybrid micromirror with chevron tunable actuator; (**b**) a closed-up view of the latching mechanism (Reprinted from Eun et al. [91] with permission of Elsevier); (**c**) schematic of the tunable resonant micromirror; and (**d**) experimental setup for testing resonant frequencies (Reprinted from Lee et al. [92] with permission of Elsevier).

**Figure 33 micromachines-13-00429-f033:**
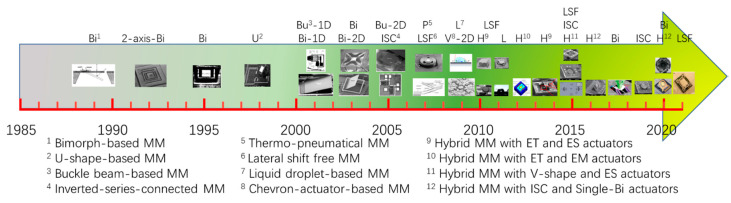
Timeline of the development of electrothermal micromirrors.

**Table 1 micromachines-13-00429-t001:** Material properties and the optimal thickness ratios for different bimorph material combinations.

Layer 1				Layer 2				Optimal Thickness Ratio
Material-1	*E*_1_ (GPa)	*a*_1_ (10^−6^K^−1^)	Melting (°C)	Material-2	*E*_2_ (GPa)	*a*_2_ (10^−6^K^−1^)	Melting (°C)	
SiO_2_	70	0.5	1710	Al	70	23.1	660	0.12, 1.7
				Au	70	14.2	1060	0.35–0.7
				Cu	120	16.5		0.13, 1
				W	411	4.5	3370	0.22
				Si	170	2.6	1410	0.344
				Poly-Si	160	2.6	1410	0.335
Si	170	2.6	1410	Al	70	23.1	--	0.32, 1.6
				Au	70	14.2	--	0.58
				Cu	120	16.5	--	0.74
				W	411	4.5	--	0.336
W	411	4.5	3370	Al	70	23.1	--	1.1
				Au	70	14.2	--	0.85
				Cu	120	16.5	--	1.1

**Table 2 micromachines-13-00429-t002:** Comparison of different types of thermally-actuated micromirrors.

Principles of Micromirrors		Stroke (μm or °)	Voltage (V)	Types	Materials	Advantages	Disadvantages	References
U-shaped		15°	9 V	1D	Poly- Si	Low voltage	Complicated	[41]
		360°-rotation	--	Rotatory	Poly- Si	Rotatory		[44]
V-shaped		6.5°	13 V	1D	Si	Reliability; Robustness	Low fill factor; Low stroke	[47]
		5.4° 5.2°	11 V	2D	Si			[47]
Buckle beams		18°	9 kHz	1D	Poly- Si	Easy to fabrication	Low fill factor	[48]
		20°	16 kHz	2D	Poly- Si			[49]
Three beams		10°	18 V 2.19 kHz	TTP	Si	Innovation	Low stroke	[50,51]
bimorph	suspend	0.3°	--	1D	SiO_2_/Al	Innovation	Low stroke	[55]
		90°	--	1D	metal/SiO_2_	Large stoke	High power consumption	[53]
	With gimbal	45°, 25°	15 V 17 V	2D	SiO_2_/Al	Large stroke; Low voltage injection	With gimble	[36]
		200 μm	6 V	TTP	SiO_2_/Al			[58]
	Origami-like	200 μm	25 V	Phase-only	NiCr/SiN	Innovation; Large stroke	Piston only	[85]
	ISC	480 μm ±30°	8 V	TTP	SiO_2_/Al	Large stroke; Low voltage injection; Lateral shift free	High power consumption	[74]
		300 μm	7 Vpp		SiO_2_/Al			[76]
		±2.8°			SiO_2_/Al			[77]
		169 μm	2.3 V		Cu/W			[80]
		32°, 22°	--		SiO_2_/Al			[83,84]
	LSF	620 μm	5.3 V		SiO_2_/Al	Large stoke; Very low voltage; Lateral shift free	High power consumption	[38,66]
		±15°	3.6 V		SiO_2_/Al			[67]
		±20°	4.5 V		SiO_2_/Al			[68]
		227 μm ±11°	0.6 V 0.8 V		Al/W			[70]
		200 μm	0.9 V		Al/W			[71]
		320 μm ±18°	3 V		Cu/W			[69]
Liquid droplet		6.5°	30 V	1D	Oil	innovation	Vulnerable	[17]
Thermal-pneumatic		12.5°	30 V	1D	PDMS/gas cavity	Large stroke	High power consumption	[18]
		80 μm	20 V	Phase-only				
		385 μm, 5°	35 V	TTP				[88]

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
