# Peer review of "(untitled)"

_micromachines, 2022, doi:10.3390/mi13030429_

Round 1

Reviewer 1 Report

In line 130, I think "increase" should be "decrease", because when the temperature of a bimorph is lowered, the bimorph will bend towards the high CTE material to minimize the internal energy stored by the stress.

Author Response

Point 1: In line 130, I think "increase" should be "decrease", because when the temperature of a bimorph is lowered, the bimorph will bend towards the high CTE material to minimize the internal energy stored by the stress. 

Response 1: Thank you for your comments. You are so right. There are something wrong to describe the bent direction when elevating the temperatue on bimorph in line 130 and in Figure 6(b). In order to express the bent orentation after incresing the temperature, I changed the ‘higher CTE’ to ‘lower CTE’ in line 130 and adjusted the sequence of ‘material 1’ and ‘material 2’ in Figure 6(a). Moreover, I also added the condition that the CTE α2 is larger than the CET α1 in Eq.(2).

Reviewer 2 Report

The authors‘ group has very extensive experience in developing electrothermal micromirrors. This review paper gives readers (even beginners) a very good database and reference to learn and know the field of electrothermal micro mirrors.   The reviewer believe this paper should be published in the journal as early as possible.

Author Response

Point 1: The authors‘ group has very extensive experience in developing electrothermal micromirrors. This review paper gives readers (even beginners) a very good database and reference to learn and know the field of electrothermal micro mirrors. The reviewer believe this paper should be published in the journal as early as possible. 

Response 1: Thank you very much for your comments.
